# DeNAV: Decentralized Self-Supervised Learning with a Training Navigator

## Abstract

Current Federated Self-Supervised Learning (FSSL) methods can achieve effective learning on edge devices with unlabeled data. However, in realistic settings, it is not easy to ensure that distributed clients at a large scale can efficiently communicate with a central server. In this work, we study an essential scenario of Decentralized Self-Supervised Learning (DSSL) based on decentralized communications. It is a highly challenging scenario where only unlabeled data is used during the pre-training stage, and the communication between clients involves only model parameters without data sharing. We propose a novel method to tackle the problems, which we refer to as Decentralized Navigator (DeNAV). DeNAV utilizes a lightweight pre-training model, namely the One-Block Masked Autoencoder, with a training navigator to evaluate selection scores for the connected clients and plan the training route based on these scores, eliminating the reliance on server aggregation in federated learning. Comprehensive experimental validation demonstrates that DeNAV surpasses the most advanced FSSL and Gossip Learning methods in terms of accuracy and communication costs.

## 1 Introduction

Distributed learning (Liu et al., 2022), in which multiple clients collaboratively learn a global model through coordinated communication, generally falls into two types: Server-Client and Client-Only architectures, as illustrated in Figure 1. Federated learning, a typical server-client architecture where the server facilitates global model training by aggregating locally learned model parameters, has been an active research topic over the past few years (Zhuang et al., 2021a;b; Lubana et al., 2022; McMahan et al., 2017; Zhang et al., 2023; Jeong et al., 2020). However, in real-world settings, not all clients can connect to the server; instead, there is more inter-client communication. Such decentralized learning offers numerous advantages in addressing realistic issues such as data privacy and security. For example, inter-communication between devices (Clients) in a smart home is generally endorsed to users, while they are unwilling to link private devices directly to an external service provider (Server), suggesting a preference for system improvement within the local area network.

Existing decentralized learning approaches handle these problems by aggregating neighbor-learned model parameters in local clients (Tang et al., 2020). A common limitation is that they only consider supervised learning settings, assuming fully labeled local data. However, the assumption is not realistic for real-world applications. Suppose that we perform on-device decentralized learning; the users may not want to spend their time and effort annotating the data, and the participation rate across the users may largely differ. Another standard limitation is that they consider different communication settings between clients, where each client communicates with all other clients in the All-Reduced system (Chu et al., 2017), and each client communicates with all neighbors in the Gossip Learning system (Chu et al., 2017; Kempe et al., 2003). However, high communication costs may result in increased power consumption for mobile devices, and more frequent communication will increase the potential risk of data interception or unauthorized access. Thus, in many realistic scenarios for decentralized learning, local data will be mostly unlabeled, and the communication cost can not be significant. This leads to practical problems of decentralized learning with a deficiency of labels and lower communication, namely Decentralized Self-Supervised Learning (DSSL).

To address these problems, we propose a novel training framework called Decentralized Navigator (DeNAV). DeNAV employs a training navigator to determine the training client for each pre-training

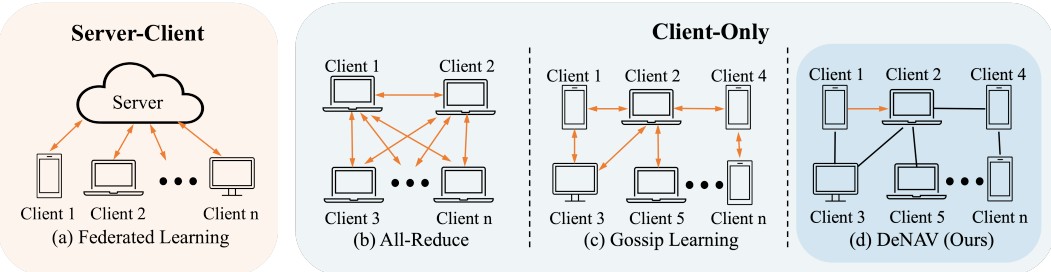

Figure 1: **Different system architectures for Distributed Learning. (a) Federated Learning**: A subset of clients train the global model from the server and send their model updates to the server. The server then aggregates the updates of all these clients. **(b) All-Reduce**: Every client trains a model and communicates with all other clients to aggregate model updates. **(c) Gossip Learning**: Every client trains a model and communicates with all neighbors to aggregate model updates. **(d) DeNAV (Ours)**: A client trains a model and communicates with one of its neighbors to transmit the model update. The client receiving the model will continue the training.

step to achieve good training performance. The navigator calculates the selection scores for clients, taking into account the data volume, computing resources, and the selection history of each client. The client with the highest score will usually be selected for training, but if multiple clients share the highest score, the navigator will recursively compare more clients to select the optimal one of the equal-scored clients. Besides, DeNAV offers acceptable communication and training costs compared to existing decentralized learning methods. As shown in Figure 1, each step of pre-training is performed on only one idle client in the network, and the training client will then communicate with only one neighboring client to transmit the model updates. The pre-training model used in DeNAV is a lightweight masked autoencoder where both the encoder and decoder contain a single transformer block. After pre-training, each client can create a transformer backbone of varying depths using the local one-block masked autoencoder and then fine-tune the transformer backbone for downstream tasks with a small amount of labeled data.

In summary, the main contributions of the paper are shown below:

- We introduce the scenario of decentralized self-supervised learning (DSSL) and rigorously prove that the pre-training performance of autoencoder depends mainly on the amount rather than the class distribution of data from the participating clients. (Section 3)
- We propose a training framework, DeNAV, for DSSL to pre-train a one-block masked autoencoder in edge devices with efficient computation and communication. (Section 4)
- We design a training navigator for DSSL and utilize it in DeNAV to achieve effective training performance. (Section 5)
- We conduct comprehensive experiments to evaluate DeNAV compared to FSSL and Gossip Learning. The results show that DeNAV achieves better model performance than FSSL and uses less computation and communication than Gossip Learning. (Section 6)

## 2 RELATED WORK

**Self-Supervised Learning and Transformer.** Since the transformer was proposed (Vaswani et al., 2017), this class of models has been trained using self-supervised learning. When transformers (Dosovitskiy et al., 2020; Liu et al., 2021) emerged in the field of computer vision, corresponding self-supervised methods also emerged, such as BeiT (Bao et al., 2021), DeiT (Touvron et al., 2021), and MAE (He et al., 2022). During the training of MAE (Masked Autoencoder), 75% of image patches are randomly masked, and an autoencoder is trained to reconstruct the masked portion. Subsequently, the encoder is fine-tuned for downstream tasks. The significant advantage of MAE is its ability to reduce memory consumption. Similarly, ALBERT (Lan et al., 2019) introduced a trick called cross-layer parameter sharing. By sharing all parameters across transformer model layers, the training overhead is drastically reduced without significantly lossing performance.

**Federated Self-Supervised Learning.** Federated learning is a collaborative training framework where there are multiple clients and a central server connected to all clients (Zhang et al., 2021; Dong et al., 2022; Liu et al., 2023; Jeong et al., 2020). Federated self-supervised learning (FSSL) combines federated learning with self-supervised learning, aiming at utilizing the unlabeled data on clients for training. Recent advancements in FSSL (Zhuang et al., 2021a;b; Lubana et al., 2022) have been comparable with federated learning methods using labeled data (McMahan et al., 2017; Zhang et al., 2023).

**Decentralized Learning.** Like federated learning, decentralized learning can utilize local client data to train models. However, decentralized learning is less popular than federated learning because existing decentralized frameworks (All-Reduce and Gossip Learning) are not communication-efficient (Tang et al., 2020). As shown in Figure 1, after each client in the network completes local training, All-Reduce requires them to transmit their model updates to all other clients (Chu et al., 2017), while Gossip Learning requires them to transmit their model updates to all their neighbors (Kempe et al., 2003; Hegedűs et al., 2019).

## 3 Theoretical Foundations for DSSL

The state-of-the-art FSSL methods employ contrastive learning (Chen et al., 2020; Chen & He, 2021; Grill et al., 2020) to pre-train convolutional neural networks. However, contrastive learning is unsuitable for DSSL because there is no central server in DSSL to mitigate the adverse effects of non-independent and identically distributed (non-IID) data on training. We thus conducted mathematical modeling and discovered that generative self-supervised methods using autoencoder are suitable for DSSL because their pre-training performance depends mainly on the data volume.

### 3.1 Preliminaries

**DSSL Scenario.** The scenario of DSSL consists of a network of $M$ clients, where the unlabeled training data $X_m$ are stored in client $m \in M := 1, \ldots, |M|$. There are no isolated nodes among these clients, meaning that communication can be initiated from any client, and after successive hops, all the other clients can be accessed. Depending on the demand, $C$ models will be simultaneously trained for $S$ steps in this network. Correspondingly, in each pre-training step, a subset of clients $P_s \subset M$ simultaneously training these models. At the beginning of the step $s = 1, \ldots, S$, the client $m$ checks if it has received a model from another client. If so, the client $m$ aggregates the received model $\theta_{s-1}$ with the local model $\theta_m$ to update $\theta_{s-1}$. The client $m$ then conducts a local training with local data of size $n_m = |X_m|$ for $K$ iterations to update the model from $\theta_{s-1}$ to $\theta_s$. After local training, a copy of the model $\theta_s$ is saved on the client $m$ to update $\theta_m$. Next, client $m$ determines all possible communication targets $P_m$ from the remaining clients in the network and selects a client $p_m$ from $P_m$ to send the model $\theta_s$. Therefore, the subset of clients responsible for local training of the models can be updated by $P_{s+1} = \bigcup_{m \in P_s} p_m$. After $S$ pre-training steps, the local model $\theta_m$ can be fine-tuned for downstream tasks.

**Self-Supervised Learning using Autoencoder.** Autoencoders are widely used in self-supervised learning (Bao et al., 2021; He et al., 2022; Xie et al., 2022). The training involves an unlabeled dataset, denoted as $X = \{x_1, x_2, \ldots, x_n\}$, and an autoencoder, which consists of an encoder $h(x) = z$ and a decoder $g(z) = x$. The autoencoder learns features from the unlabeled data by data reconstruction, represented as a pairwise encoder-decoder relationship $\hat{x} = g(h(x))$. To quantify the difference between the original data $x$ and the reconstructed data $\hat{x}$, a loss function $l(x, \hat{x})$ is used. However, due to artificial corruption operations such as adding noise and masking, the input $x$ has been degraded to $\tilde{x}$. Thus, during training, the parameters of the autoencoder, denoted as $\theta$, are optimized by solving the problem: $\theta = \arg\min_\theta \frac{1}{n} \sum_{i=1}^{n} l(x_i, g(h(\tilde{x}_i)); \theta)$.

### 3.2 Impact of Data Volume on DSSL using Autoencoder

With the above mathematical definitions, when DSSL uses autoencoder as the pre-training model, the total loss produced in each step can be denoted as $\ell_s = \frac{1}{C} \sum_m \frac{1}{n_m} \sum_{i=1}^{n_m} l(x_i, g(h(\tilde{x}_i)); \theta_s^m)$,

where $m \in P_s$, $s \leq S$ and $x_i \in X_m$. Reconstruction of the input image is represented by the joint relation of the $h(\cdot)$ and $g(\cdot)$ operators. It should be noted that when the input is an image where all

pixels are 0, the output of the encoder is a tensor that tends to zero, i.e., $h(0) \to 0$. Similarly, when the input is a zero tensor, the output of the decoder is a reconstructed image with all zero pixels, that is, $g(0) = 0$. As a result, there is a zero reference point for the discussion of the encoder and decoder operators. Based on these observations, the following proposition is derived.

**Proposition 1.** *There exists a linear equivalent mapping with $W_h$ to the approximate encoder $h(\cdot)$ and a linear equivalent mapping with $W_g$ to the approximate decoder $g(\cdot)$.*

*Proof.* Due to page limits, the detailed proof of Proposition 1 is provided in **Appendix A.2.1**. □

**Remark 1.** *It is important to note that this linear representation may have limitations since it lacks higher-order feature representations and residual terms. Despite this, our main focus is on understanding the nature of the mapping. It is used as a means to simplify the problem.*

For local training on the client $m$, the approximate linear equivalence mapping allows simplification of the model. Consequently, the approximate encoder $h(\cdot)$ can be represented as a linear mapping $W_{h_m}$. Similarly, the decoder $g(\cdot)$ can be expressed as another linear mapping $W_{g_m}$, resulting in the equation $\hat{x} = g_m(h_m(\tilde{x})) \approx W_{g_m} W_{h_m} \tilde{x} = W_m x$. Then, the following theorem is obtained.

**Theorem 1.** *In each pre-training step of DSSL, the approximate optimal solution for the model over the $C$ participating clients can be obtained by $W_A^*$, in which*

$$W_A^* = \bar{X}\tilde{X}^T(\tilde{X}\tilde{X}^T)^{-1} \tag{1}$$

*where*

$$
\tilde{X} = \begin{bmatrix} \tilde{x}_{1,1} & \cdots & \tilde{x}_{1,C} \\ \vdots & \tilde{x}_{i,j} & \vdots \\ \tilde{x}_{n^{\max},1} & \cdots & \tilde{x}_{n^{\max},C} \end{bmatrix}_{n^{\max} \times C} \quad X = \begin{bmatrix} x_{1,1} & \cdots & x_{1,C} \\ \vdots & x_{i,j} & \vdots \\ x_{n^{\max},1} & \cdots & x_{n^{\max},C} \end{bmatrix}_{n^{\max} \times C}
$$
$$
n^{\max} = \max\{n_m \mid m \in P_s\}, \quad \tilde{X}_{i,j}, X_{i,j} = \begin{cases} 0, & \text{if } i > n_m \\ \tilde{x}_{i,j}, x_{i,j}, & \text{otherwise} \end{cases} \tag{2}
$$

*Proof.* Due to page limits, the detailed proof of Theorem 1 is provided in **Appendix A.2.2**. □

**Corollary 1.** *In each pre-training step of DSSL, the training performance depends mainly on the amount of data in the selected clients.*

**Remark 2.** *Theorem 1 reveals that the approximate optimal solution $W_A^*$ is related to $\tilde{X}$ and $X$. In each step of pre-training, since the data $X_m$ in each selected client $m$ is invariant and random data corruption is performed in $X_m$ to generate $\tilde{X}_m$, the controllable factors are $C$, $n_m$, and $n^{\max}$, where $C$ is kept constant each step, while $n^{\max}$ depends on $n_m$. If the selected clients have large data volumes, both $n_m$ and $n^{\max}$ will increase. Consequently, the size of the matrix $W_A^*$ (which is $n^{\max} \times n^{\max}$) will increase, and the matrix $W_A^*$ will be less sparse. Thus, the approximate optimal solution $W_A^*$ of the global model obtained from training will be closer to the ground-truth mapping from input to output, enhancing training performance.*

## 4  DENAV OVERVIEW

Based on the theoretical foundation described above, we propose the Decentralized Navigator (De-NAV) to solve the DSSL problem. In this section, we provide an overview of the pre-training phase and the fine-tuning phase of DeNAV.

**Pre-training.** In the scenario of DSSL, clients' computing resources are limited so that large autoencoders with multiple transformer blocks for both the encoder and decoder can not be deployed. Therefore, in DeNAV, the pre-training model is a one-block masked autoencoder. To reduce computational overhead on clients, we minimize both the encoder and decoder to contain only one transformer block and mask 75% of image patches out of the local data during the local training.

Before pre-training begins, an idle client is randomly selected from the network for the first pre-training step. A one-block masked autoencoder is initialized on this client along with its associated state log, which records information about the training state. Subsequently, a training procedure

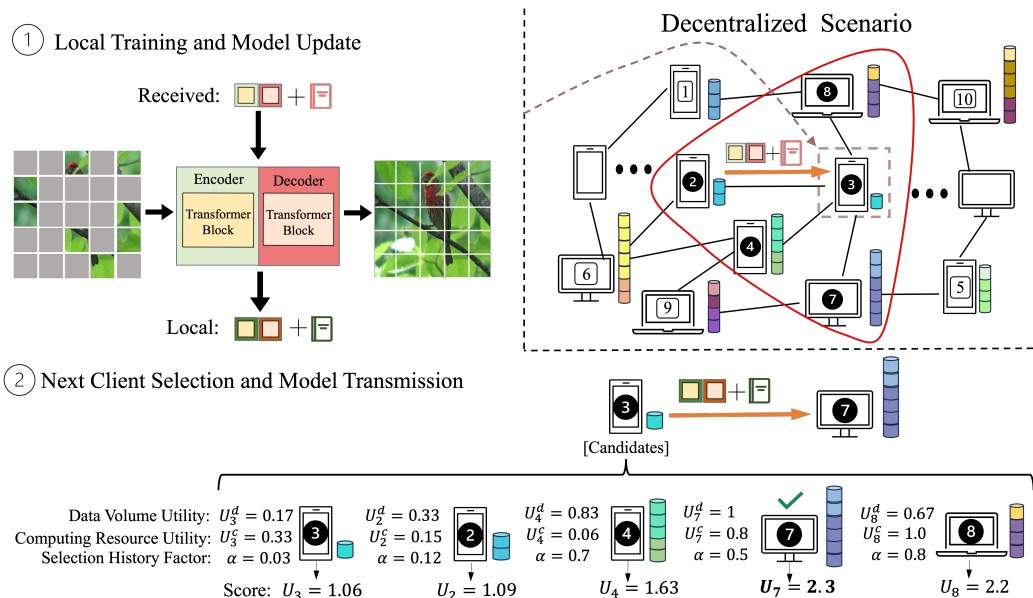

Figure 2: **Illustration of training procedure of DeNAV.** The DSSL scenario is a client-only architecture where clients vary in terms of data classes, data volume, and computing resources. At the beginning of each pre-training step, a client receives the one-block masked autoencoder and its state log from a neighbor. During pre-training: ① The client trains the received model using local unlabeled data, updates the received state log, and creates a local copy of both. ② The training navigator calculates the selection score of each candidate to find the client for the next pre-training step. The latest model and state log are transmitted from the current client to the selected client.

is repeated throughout the pre-training. As illustrated in Figure 2, the operations in this training procedure are: ① *Local Training and Model Update*: A client receives the pre-training model and the state log of this model from its neighbor or model initialization. The received model is trained using unlabeled local data, and the state log is updated at the end of the local training. Then, a copy of the latest model and state log is saved on the client. If a local model and a local state log exist, the local copy will be updated. ② *Next Client Selection and Model Transmission*: A training navigator is utilized to determine the training client for the next step of pre-training. The training navigator calculates the selection score for each candidate (consisting of the training client and the neighbors of the training client) and selects the candidate with the highest selection score. Subsequently, communication is initiated from the current client to the selected client, transmitting the latest model and the state log of this model.

Furthermore, DeNAV can be deployed in DSSL scenarios with parallel training for better performance, but this increases training and communication costs. Due to page limits, the details of DeNAV with parallel training can be found in **Appendix A.4.**

**Fine-tuning.** After pre-training, each client has a local copy of the one-block masked autoencoder. The decoder of this model is discarded, and its encoder can be utilized for fine-tuning at any time. During fine-tuning, a client first averages the local encoder block with those from neighbors to obtain a pre-trained transformer block. Based on the computing resources, the client creates a trainable large transformer backbone with multiple transformer blocks by sharing the parameters of the pre-trained transformer block. Lastly, the transformer backbone is fine-tuned using a small amount of labeled data from the local client.

## 5   THE TRAINING NAVIGATOR IN DeNAV

Due to variations among clients, selecting which client for the next step of pre-training significantly affects the training performance of DSSL. The above training procedure shows that DeNAV em-

ploys a training navigator to determine the optimal training route of the model in the network. The navigator does two main tasks: evaluating the selection scores of candidates and selecting the next training client based on these scores.

## 5.1 Client Selection Score Formulation

The objective of the training navigator is to find a sweet spot in the trade-off between the training performance and the training efficiency by associating each client with its selection score. Achieving this goal leads to four challenges: 1. How to determine which client's data would help improve the training performance the most without compromising privacy? 2. How to take into account optimization of training efficiency while optimizing training performance? 3. How to balance between exploring new clients and continuing to exploit the clients that have been selected for maximum gain? 4. How to ensure that all clients in the network have the opportunity to participate in the model pre-training? To address all these challenges, the navigator evaluates a client's local data, computing resources, and selection history.

**Data Volume Utility.** Corollary 1 shows that the training performance depends mainly on the data volume in the selected clients. Therefore, the importance of the data on the client $m$ for training should be quantified by its data volume $n_m$. Intuitively, this utility is formulated as $U_m^d = \frac{n_m}{\sum\limits_{m \in P_m} n_m}$, where $P_m$ represents the candidates of the client $m$. However, this formulation has limitations, as the sum of data volumes can be excessively large compared to individual data volumes, resulting in an insignificant difference in the data volume utilities between clients with more data and those with less data. Therefore, we define the data volume utility $U_m^d$ as:

$$U_m^d = \frac{n_m}{\max\{n_m \mid m \in P_m\}} \tag{3}$$

where the denominator is the maximum data volume of all clients. Since the data volume utility only focuses on the amount of data instead of raw data, the privacy of client $m$ is preserved.

**Computing Resource Utility.** In real-world settings, clients with more computing resources can train models at a larger batch size and finish the training in less time than clients with fewer computing resources, demonstrating greater training efficiency. Thus, the training efficiency of the client $m$ is measured by the computing resource utility $U_m^c$, formulated as:

$$U_m^c = \left(\frac{\min\{t_m \mid m \in P_m\}}{t_m}\right)^{\mathbb{1}(L(m)>0)} \tag{4}$$

where $t_m$ is the time spent by the client $m$ on its last local training, $L(m)$ denotes the last step that the client $m$ selected for local training, and $\mathbb{1}(x)$ is an indicator function that returns 1 if $x$ is true and 0 otherwise. The maximum computing resource utility is allocated to never-selected clients to encourage more client engagement and determine their local training time.

**Selection History Factor.** Simply integrating the two utilities is insufficient to formulate an ideal client selection score, as the navigator tends to exploit the already identified excellent clients rather than explore new ones. This repeated selection of the same group of clients can lead to diminishing training rewards as the model overfits their local data. Moreover, in real scenarios, clients' data volume and computing resources may change during pre-training, making previously identified superior clients no longer advantageous. To harmonize the trade-off between exploration and exploitation during training, the selection history factor $\alpha$ is introduced, formulated as:

$$\alpha = \left(\frac{s - L(m)}{S}\right)^{\mathbb{1}(L(m)>0)} \tag{5}$$

where $s$ is the current pre-training step, and $S$ is the total number of pre-training steps. The inclusion of $\alpha$ gradually increases the selection scores of clients which have not been selected for a long time, allowing well-qualified clients with sufficient scores since their last training to be reselected.

**Total Selection Score.** The navigator combines the two utilities and a factor into the client selection score. First, the data volume utility should be associated with the computing resource utility as $U_m^d \times U_m^c$ in order to yield good training results while optimizing the training efficiency. Furthermore, since we prioritize the training performance over training efficiency, the association of $\alpha$ and two

utilities is formulated as $U_m^d \times (\alpha + U_m^c)$ instead of $(\alpha + U_m^d) \times U_m^c$. Lastly, to ensure the utilization of all clients' data during training, the client selection score $m$ is formulated as:

$$U_m = (U_m^d \times (\alpha + U_m^c) + 1)^{\mathbb{1}(Z(m)<Z)}, Z \geq \frac{S}{|M|} \qquad (6)$$

where $Z(m)$ is the count of the previous selections of client $m$, and $Z$ is the maximum allowed selections per client. If a client reaches the maximum selection count, its score is adjusted to a minimum of 1, allowing other clients the chance for selection. Besides, the upper limit of client selection should depend on the ratio of the total training steps $S$ to the number of clients $|M|$ in the network. If $|M|$ remains constant but $S$ increases, each client can be selected more times while ensuring complete use of data for model training.

## 5.2 NEXT CLIENT SELECTION

With the selection scores of candidates, the training navigator determines the client for the next step of pre-training. The simplest approach is to select the client with the highest score, but multiple clients may share the highest score. For instance, when the selection score of each candidate is 1 due to frequent selection, randomly selecting any client may not be optimal. Therefore, the navigator will use different client selection strategies depending on whether this situation occurs.

Initially, the training navigator determines the candidate set $P_m$ for the training client $m$ and calculates the selection scores for all clients in $P_m$ using the formula 6. If a single client in $P_m$ has the highest selection score, it is selected as the communication target of client $m$ and the training client for the next pre-training step. However, if multiple clients in $P_m$ share the highest selection score, the training navigator iteratively searches the neighbors of these candidates and compares their neighbors' scores to determine which candidate to select. The recursion starts with the neighbors whose communication hops, $d$, equal to 1. If multiple neighbors also share the highest selection score, the recursion is continued to neighbors that require $d + 1$ communication hops until the tie score is broken or the stop condition is satisfied. Regardless of the search results, a client from $P_{m,d-1}$ with the highest selection score is chosen. In the case that multiple such clients exist in $P_{m,d-1}$, one is randomly chosen. The navigator then backtracks the recursive search to identify the corresponding client $p_m$ in $P_m$ for the one chosen in $P_{m,d-1}$, and selects this client from $P_m$ as the communication target of the client $m$. Due to page limits, detailed client selection performed by the training navigator is provided in **Appendix A.3**.

# 6 EXPERIMENTS

## 6.1 EXPERIMENT SETUP

**Dataset.** In our experiments, we utilized various datasets, including ImageNet (Deng et al., 2009), Mini-ImageNet, and Mini-INAT2021, which are datasets with large-size images, and CIFAR10 and CIFAR100 (Krizhevsky et al., 2009), which are datasets with small-size images. Both ImageNet and Mini-ImageNet have 224x224 images. Mini-ImageNet is a dataset with 60000 images in 100 classes, selected from ImageNet through the methodology detailed in (Vinyals et al., 2016). Mini-INAT2021 is a reduced version of the INaturalist-2021 dataset (ina, 2021) containing 10000 classes and a maximum image size of 800 pixels. CIFAR10 and CIFAR100 each include 60,000 32x32 color images, with CIFAR10 having 10 classes and CIFAR100 having 100 classes.

**System Settings.** We evaluated the performance of DeNAV and baselines on a simulated Internet of Things (IoT) network. This network is initialized using the Erdős-Rényi model (ERDdS & R&wi, 1959), where the total number of clients is 100 and the network connectivity is 0.15. For FSSL baselines, we assumed a server connected to all clients in this network. During experiments, the Mini-ImageNet dataset is used for pre-training, and the remaining datasets are used for downstream evaluation. The Mini-ImageNet dataset is partitioned according to the number of clients. The amount of data on each client is randomly assigned. If it is assumed that the local data is IID, then each client will hold images of all categories. However, if it is assumed that the data follows a non-IID distribution, the dataset will be divided by sampling the category priors of the Dirichlet distribution (Hsu et al., 2019) so that each client will have images of a few categories. Additionally, we allocate random computing resources with a scale rank from 1 to $H$ to each client and set the

local training time of a client to be affected by its computing resources. The impact is formulated as $t_m = \hat{t}_m / (1 + \frac{h_m - 1}{H - 1})$, where $t_m$ is the training time updated in the model state log, $\hat{t}_m$ is the actual time consumed by the client $m$ to train the model, and $h_m$ is the scale of computing resources on the client $m$. (Due to page limits, detailed system settings are provided in **Appendix A.5**.)

## 6.2 COMPARISON WITH FEDERATED LEARNING BASELINES

To comprehensively assess the effectiveness of our proposed DeNAV framework, the following state-of-the-art FSSL benchmarks were compared: **1) Fed-SimSiam**: a naive combination of SimSiam (Chen & He, 2021) and Federated Learning. **2) Fed-SimCLR**: a naive combination of SimCLR (Chen et al., 2020) and Federated Learning. **3) FedU** (Zhuang et al., 2021a): Using the divergence-aware predictor module for dynamic updates within the self-supervised BYOL network (Grill et al., 2020). **4) FedEMA** (Zhuang et al., 2021b): Employing EMA of the global model to adaptively update online networks. **5) Orchestra** (Lubana et al., 2022): Combining clustering algorithms with Federated Learning for better model aggregation.

Table 1: Comparison of DeNAV with FSSL baselines in terms of model parameters and GFLOPs. The size of the input image is 224x224.

|              | Fed-SimSiam | Fed-SimCLR | FedU   | FedEMA | Orchestra | DeNAV (pre-train) | DeNAV (Downstream) |
|--------------|-------------|------------|--------|--------|-----------|-------------------|--------------------|
| Model Params | 12.03M      | 11.70M     | 38.47M | 38.47M | 11.84M    | **11.62M**        | 39.97M             |
| GFLOPs       | 3.65        | 3.65       | 7.40   | 7.40   | 7.31      | **1.23**          | 7.39               |

Since the pre-training model is not a convolutional neural network and clients can construct downstream models of varying depths using a pre-trained one-block masked autoencoder, we first compare the computational overhead (measured in GFLOPS) and model size (quantified by the number of model parameters) of DeNAV and the baselines during pre-training and fine-tuning to ensure a fair comparison. As shown in Table 1, DeNAV, taking the experimental settings in the Appendix, is comparable to the FSSL baselines in terms of computational overhead and model size. Besides, DeNAV not only achieves pre-training of transformers but also has a lower training cost than the baselines during pre-training.

Table 2: Accuracy (%) on the CIFAR10, CIFAR100, ImageNet, and Mini-INAT datasets following IID or non-IID data distribution. For pre-training, the local training epochs were set to 10. For the downstream evaluation, each model was fine-tuned for 100 epochs. The experimental results show the mean and standard deviation of three trials.

| Method | CIFAR10 | | CIFAR100 | | ImageNet | | Mini-INAT | |
|--------|---------|---------|----------|---------|----------|---------|-----------|---------|
|        | IID     | non-IID | IID      | non-IID | IID      | non-IID | IID       | non-IID |
| Fed-SimSiam(%) | 89.91 | 89.58 | 68.52 | 71.46 | 65.26 | 64.87 | 32.57 | 37.43 |
|                | ±0.17 | ±0.21 | ±0.31 | ±0.44 | ±0.42 | ±0.20 | ±0.13 | ±0.30 |
| Fed-SimCLR(%)  | 89.54 | 90.39 | 67.20 | 71.24 | 65.47 | 65.32 | 37.70 | 37.60 |
|                | ±0.20 | ±0.22 | ±0.09 | ±0.12 | ±0.13 | ±0.41 | ±0.09 | ±0.43 |
| FedU(%)        | 77.43 | 72.02 | 40.40 | 38.44 | 65.34 | 65.34 | 37.88 | 37.61 |
|                | ±0.12 | ±0.19 | ±0.02 | ±0.18 | ±0.20 | ±0.37 | ±0.24 | ±0.46 |
| FedEMA(%)      | 70.73 | 71.00 | 40.78 | 41.13 | 65.24 | 65.35 | 38.40 | 37.43 |
|                | ±0.08 | ±0.06 | ±0.37 | ±0.06 | ±0.03 | ±0.14 | ±0.49 | ±0.27 |
| Orchestra(%)   | 88.87 | 90.66 | 72.11 | 72.27 | 65.02 | 66.50 | 38.74 | 39.23 |
|                | ±0.14 | ±0.19 | ±0.20 | ±0.09 | ±0.35 | ±0.21 | ±0.09 | ±0.33 |
| DeNAV(%)       | **91.12** | **91.00** | **74.50** | **73.89** | **77.49** | **77.62** | **46.38** | **44.98** |
|                | ±0.21 | ±0.26 | ±0.10 | ±0.23 | ±0.21 | ±0.05 | ±0.27 | ±0.19 |

Then, we pre-trained models using DeNAV and FSSL baselines and collected their fine-tuning accuracy on downstream datasets. In this experiment, the number of local training epochs is 10 instead of the default 5 during pre-training. Table 2 demonstrates that DeNAV achieves superior results not only on small image datasets such as CIFAR10 and CIFAR100 but also on large-scale datasets such as ImageNet and Mini-INAT, although there is only a single-client training model in each pre-training step and there is no model aggregation in DeNAV compared to the baselines. The improvement in fine-tuning accuracy on large-scale datasets is particularly significant, with an improvement of approximately 12% on ImageNet and 5% on Mini-INAT.

Table 3: Comparison of DeNAV and Gossip Learning. "Epochs" represents the total number of local training epochs for all training clients. "Communication Cost" indicates the total number of client-to-client communications occurring in a network with 100 clients and 0.15 connectivity.

|  | CIFAR10(%) | CIFAR100(%) | Communication Cost |
|---|---|---|---|
| DeNAV ($Epochs = 5000$) | **92.80** | **75.62** | **1000** |
| Gossip Learning ($Epochs = 5000$) | 90.52 | 72.34 | 14850 |
| DeNAV ($Epochs = 15000$) | **93.44** | **77.30** | **3000** |
| Gossip Learning ($Epochs = 15000$) | 90.30 | 72.52 | 44550 |
| DeNAV ($Epochs = 25000$) | **93.88** | **77.94** | **5000** |
| Gossip Learning ($Epochs = 25000$) | 92.91 | 75.71 | 74250 |

## 6.3 COMPARISON WITH GOSSIP LEARNING

DeNAV was also compared to other decentralized learning approaches. Since All-Reduce is only suitable for high-performance computing clusters, a comparison was made between DeNAV and Gossip Learning, which can also be applied to edge devices. Before the experiment, we implemented the baseline by referring to the Gossip Learning architecture described in Tang's paper (Tang et al., 2020). In our version of Gossip Learning, each client trains a one-block masked autoencoder with local data and aggregates the local model with the first received model from neighbors. Table 3 shows that DeNAV has better fine-tuning accuracy and incurs much lower overall communication costs during pre-training than Gossip Learning.

Table 4: Analysis on Client Selection.

| Formula | CIFAR10(%) | CIFAR100(%) | Pre-train Time (min) |
|---|---|---|---|
| Random Selection | 90.28 | 71.70 | 33.7 |
| Formula w/o data volume utility $U_m^d$ | 88.54 | 71.99 | **28.8** |
| Formula w/o computing resource utility $U_m^t$ | 90.65 | 73.46 | 37.6 |
| Formula w/o selection history factor $\alpha$ | 90.23 | 73.51 | 38.9 |
| Our Formula | **90.96** | **73.70** | 36.8 |

## 6.4 FURTHER ANALYSIS

**Client Selection Ablation Study.** One of the important contributions of this paper is the proposal of a training navigator for DSSL scenarios. We further conducted an ablation study to analyze the impact of employing the training navigator on the training performance of DeNAV. The performance breakdown provided in Table 4 indicates that selecting clients using the training navigator enhances training, and each component of the client evaluation formula plays a constructive role in the improvement. Specifically, the data volume utility $U_m^d$ primarily drives training improvements, while the computational resource utility $U_m^c$ reduces time while improving training performance. The selection history factor $\alpha$ has the smallest impact on training but also produces a positive effect.

**Hyper-parameter Impact Analysis.** We also conducted experiments to comprehensively analyze the impact of some hyper-parameters from the system settings on the training of DeNAV. Due to page limits, the experimental results can be found in Appendix A.6.

## 7 CONCLUSION

While FSSL has achieved effective training using unlabeled data on edge devices while preserving privacy, the training scenario required for FSSL is difficult to deploy under realistic conditions. In this paper, we introduce the DSSL scenario, which is more practical than FSSL, and propose a novel training framework called DeNAV to tackle the challenges of DSSL. DeNAV pre-trains a lightweight model, the One-Block Masked Autoencoder, equipped with a training navigator to assess selection scores for clients and plan the training route in the network of clients based on their scores. Through extensive experimental validation, DeNAV demonstrates superior performance compared to existing distributed learning mechanisms and lower communication cost compared to other decentralized learning methods such as Gossip Learning.

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

# A    APPENDIX

## A.1    MATH NOTATION

### System Architecture Related

| | |
|---|---|
| $m$ | The client |
| $M$ | The set of all clients |
| $x$ | An unlabeled image |
| $X$ | The set of unlabeled images |
| $s$ | The current step of pre-training |
| $S$ | The total number of pre-training steps |
| $C$ | The number of simultaneously pre-trained models |
| $P_s$ | The set of training clients in the pre-training step $s$ |
| $P_m$ | The set of the communication candidates for the client $m$ |
| $p_m$ | The communication target of the client $m$ |
| $d$ | The number of communication hops between two clients |
| $P_{m,d}$ | The set of the clients that are $d$ hops away from the client $m$ |
| $\theta$ | The pre-trained model |
| $\Theta$ | The set of pre-trained models |
| $n$ | The data volume |
| $\psi$ | The state log of the pre-trained model |
| $\Psi$ | The set of state logs |
| $h$ | The computing resources of a client |
| $H$ | The highest scale of computing resources |
| $\omega$ | The network connectivity |

### Pre-training Related

| | |
|---|---|
| $h(\cdot)$ | Encoding |
| $g(\cdot)$ | Decoding |
| $\tilde{x}$ | Corrupted Image |
| $\hat{x}$ | Reconstructed Image |
| $l$ | Loss |
| $\epsilon$ | A higher order infinitesimal residual |
| $W_h$ | Linear Equivalent Mapping to the encoder |
| $W_g$ | Linear Equivalent Mapping to the decoder |
| $W_A^*$ | The approximate optimal solution of the pre-trained model |
| $t$ | The time of local training |
| $\lambda$ | The staleness bound |
| $U$ | The selection score of a client |
| $U^d$ | The data volume utility |
| $U^c$ | The computing resource utility |
| $\alpha$ | The selection history factor |
| $z$ | The selection times of a client |
| $Z$ | The maximum times that a client can be selected |
| $L$ | The last step that a client is selected |

## A.2    MATHEMATICAL PROOFS

### A.2.1    PROOF FOR PROPOSITION 1

*Proof.* Expanding the nonlinear vector function $h(x)$ into a Taylor series at 0, we have

$$h(x) = h(0) + \nabla_x h(0)x + \epsilon \tag{7}$$

where $\nabla_x h(0)$ denotes the gradient of operator $h(\cdot)$ at 0 in the direction of the vector $x$, and $\epsilon$ a higher order infinitesimal residual. By neglecting the residual and letting $\nabla_x h(0)x = W_h x$, we get

$$h(x) \approx W_h x + h(0) \tag{8}$$

As $h(0) \to 0$, $h(x)$ can be represented by the mapping $W_h$. Likewise, by neglecting the residual and letting $\nabla_x g(0)x = W_g x$, we get

$$g(x) \approx W_g x + g(0) \tag{9}$$

Since $g(0) = 0$, $g(x)$ can be represented by the mapping $W_g$, thus completing the proof. $\quad\square$

### A.2.2 PROOF FOR THEOREM 1

*Proof.* With local data of size $n_m$, the corrupted input to the model on client $m$ can be formulated as

$$\tilde{X}_m = [\tilde{x}_1, \ldots, \tilde{x}_{n_m}] \tag{10}$$

The ground-truth data can be formulated as

$$X_m = [x_1, \ldots, x_{n_m}] \tag{11}$$

Then, the transformed loss function of the training with the aggregation can be formulated as

$$
\begin{aligned}
l(X_m, \hat{X}_m) &= l(X_m, g_m(h_m(\tilde{X}_m))) \\
&= \tfrac{1}{2}\|X_m - W_m \tilde{X}_m\|^2 \\
&= \tfrac{1}{2}tr\left[(X_m - W_m \tilde{X}_m)(X_m - W_k \tilde{X}_m)^T\right]
\end{aligned} \tag{12}
$$

This loss function is a convex function that can reach a minimum value when its derivative is 0. Therefore, with $\nabla_{W_m} l(X_m, \hat{X}_m) = 0$, it yields

$$
\begin{aligned}
2\nabla_{W_m} l(X_m, \hat{X}_m) &= \nabla_{W_m} tr\left[(X_m - W_m \tilde{X}_m)(X_m - W_m \tilde{X}_m)^T\right] \\
&= \nabla_{W_m} tr(X_m^T X_m - \tilde{X}_m^T W_m^T X_m - X_k^T W_m \tilde{X}_m + \tilde{X}_m^T W_m^T W_m \tilde{X}_m) \\
&= \nabla_{W_m} tr(W_m \tilde{X}_m \tilde{X}_m^T W_m^T) - 2\nabla_{W_m} tr(W_m X_m \tilde{X}_m^T) \\
&= 2W_m \tilde{X}_m \tilde{X}_m^T - 2X_m \tilde{X}_m^T = 0
\end{aligned} \tag{13}
$$

Solving the equation 13 yields $W_m^* = X_m \tilde{X}_m^T (\tilde{X}_m \tilde{X}_m^T)^{-1}$. Next, if we aggregate the input and the ground-truth data over the $C$ selected clients for each step, we have

$$
\begin{aligned}
\tilde{X} &= [\tilde{X}_{m_1}, \ldots, \tilde{X}_{m_C}]^T \\
X &= [X_{m_1}, \ldots, X_{m_C}]^T
\end{aligned} \tag{14}
$$

Since $n_m$ varies from client to client, it is necessary to append $0$ to the empty space of $\tilde{X}$ and $X$. Finally, the approximate optimal solution for the model $W_A$ can also be represented in the same way by $\tilde{X}$ and $X$, so the proof is completed. $\quad\square$

## A.3 PSEUDOCODE OF THE TRAINING NAVIGATOR

---

**Algorithm 1** The Training Navigator

---

**Input:** Current step $s$, Total steps $S$, Current participating clients $P_s$, The maximum number of selections for each client $Z$

**Output:** Participating clients of the next step $P_{s+1}$

1:   $P_{s+1} \leftarrow \emptyset$
2: **for all** client $m$ in $P_s$ **do**
3:     Initialize the candidate set $P_m$ consisting of the neighboring clients of $m$ and $m$ itself
4:     $U_m \leftarrow EvaluateValue(P_m, s, S, Z)$          ▷ Evaluating function refers to formula 6
5:     $count\_max \leftarrow FindNumOfMax(U_m, \max\{U_m\})$
6:     **if** $count\_max = 1$ **then**       ▷ Case when there is only one client with the highest score
7:       Select client $p_m$ with the highest score $\max\{U_m\}$ from $P_m$
8:     **else**            ▷ Case when there are multiple clients with the highest score
9:       $d \leftarrow 1$               ▷ $d$ is the hop distance to client $m$
10:      $P_{m,d} \leftarrow FindClientsWithMaxValue(P_m, \max\{U_m\})$
11:      $\bar{P}_m \leftarrow P_m$          ▷ $\bar{P}_m$ is the set of clients that have been checked
12:      **while** $count\_max > 1$ **do**
13:        $P_{m,d+1} \leftarrow \emptyset$
14:        **for all** client $\hat{m}$ in $P_{m,d}$ **do**
15:          Identify the neighboring clients $P_{\hat{m}}$ of $\hat{m}$
16:          $P_{m,d+1} \leftarrow P_{m,d+1} \cup P_{\hat{m}}$
17:        **end for**
18:        **if** there are clients in $P_{m,d+1}$ that are not in $\bar{P}_m$ **then**
19:          $\bar{P}_m \leftarrow \bar{P}_m \cup P_{m,d+1}$
20:          $U_m \leftarrow EvaluateValue(P_{m,d+1}, s, S, Z)$
21:          $count\_max \leftarrow FindNumOfMax(U_m, \max\{U_m\})$
22:          $P_{m,d+1} \leftarrow FindClientsWithMaxValue(P_{m,d+1}, \max\{U_m\})$
23:          $d \leftarrow d + 1$
24:          **if** $\max\{U_m\} \leq 1$ **then**
25:            break           ▷ Unable to continue when the highest score is 1
26:          **end if**
27:        **else**
28:          $d \leftarrow d + 1$
29:          break           ▷ No more unchecked neighboring clients so stop
30:        **end if**
31:      **end while**
32:      Randomly selects a client $p_{m,d-1}$ with the highest score from $P_{m,d-1}$
33:      Select client $p_m \in P_m$ on the shortest path from $m$ to $p_{m,d-1}$.
34:     **end if**
35:     $P_{s+1} \leftarrow P_{s+1} \cup p_m$
36: **end for**

---

## A.4 DeNAV with Parallel Model Training

With some modifications to the training procedure, DeNAV can also achieve parallel training of multiple models, similar to existing FSSL methods. This section describes the modified training procedure of DeNAV with parallel model training and a novel model aggregation algorithm involved in the procedure, namely staleness-aware model aggregation. Unlike federated approaches, the model aggregation during pre-training is asynchronous, and the aggregation targets are the model received by the client and multiple local models on the same client.

### A.4.1 Modified Training Procedure

Before pre-training begins, given the number of models to be trained simultaneously, an equal number of idle clients are selected from the network for the first step of pre-training. A one-block masked autoencoder is initialized on one of these clients and then synchronized with the other selected clients through communication. Subsequently, the state log corresponding to that model is generated on the same client. Preparation is completed when all clients have confirmed that they have their model and the model's state log.

During pre-training, a new training procedure consisting of the following operations is repeated for a number of pre-training steps. ① *Staleness-aware Model Aggregation*: Each client selected for local training receives the pre-training model from a neighbor or model initialization. The client then compares the received model's state log with the state log from each local model to examine if there are local models meeting the criteria for staleness-aware model aggregation. If such models exist, they are aggregated with the received model based on the information in the state logs of both parties, updating the received model.② *Local Training and Model Update*: Each participating client performs local training consistent with the original procedure. After training, each client updates the state log of the received model and compares the state log of the received model with the state log of each local model. If a local model is found by comparison to be a local copy of a received model, then the received model replaces the local model, and the received state log replaces the local state log. Otherwise, a local copy of the received model and the received state log is saved on the client. ③ *Next Client Selection and Model Transmission*: This operation is the same as the operation in the original procedure. Each participating client performs this operation.

---

**Algorithm 2** Staleness-aware Model Aggregation

---

**Input:** Received model $\theta$, Received state log $\psi$, The set of locally saved models $\Theta_m$, The set of locally saved state logs $\Psi_m$, Staleness bound $\lambda$
**Output:** The model $\theta$ for subsequent local training
1: $\Theta^{Agg} \leftarrow \{\theta\}; w^{Volume} \leftarrow \{\psi^{Volume}\}; w^{Stale} \leftarrow \{\psi^{Steps}\}; w \leftarrow \emptyset$
2: **for all** $\theta_m$ in $\Theta_m$ **do**
3:     **if** $\psi_m^{ID} \neq \psi^{ID}$ and $|\psi_m^{Steps} - \psi^{Steps}| \leq \lambda$ **then**
4:         $\Theta^{Agg} \leftarrow \Theta^{Agg} \cup \theta_m; w^{Volume} \leftarrow w^{Volume} \cup \psi_m^{Volume}; w^{Stale} \leftarrow w^{Stale} \cup \psi_m^{Steps}$
5:     **end if**
6: **end for**
7: $w^{Volume} \leftarrow SoftMax(w^{Volume}); w^{Stale} \leftarrow SoftMax(w^{Stale})$
8: **for** $i = 0, 1, \ldots, |\Theta^{Agg}| - 1$ **do**
9:     $w \leftarrow w \cup (w^{Volume}[i] \times w^{Stale}[i])$
10: **end for**
11: $w \leftarrow SoftMax(w)$
12: $\theta \leftarrow \sum_{i=1}^{|w|} w[i]\Theta^{Agg}[i]$

---

### A.4.2 Staleness-aware Model Aggregation

According to the experience of federated learning, effective model aggregation can combine the strengths of each model and leverage their collective knowledge to improve performance. Therefore, model aggregation is still used in DeNAV, but given the differences between the architectures, DeNAV employs asynchronous model aggregation between the model received by the client and the local models on the same client. We notice that if a client is selected again after a long interval, the local models saved on the client are much more stale in training than the received model. Simply

aggregating the local models and the received model using average weights will not output a better model than the received model. To address this issue, we take the staleness of each model into account during model aggregation.

Algorithm 2 shows the staleness-aware model aggregation employed in DeNAV. The algorithm starts by initializing the list of aggregated models $\Theta^{Agg}$, the data volume weight list $w^{Volume}$, the staleness weight list $w^{Stale}$, and the aggregation weight list $w$ (line 1). For each local model on client $m$, its state log $\psi_m$ is compared with the state log of the received model $\psi$. If the local model $\theta_m$ isn't a previous local copy of the received model $\theta$ and the step interval between the two models falls within the staleness bound $\lambda$, the local model $\theta_m$ is included in $\Theta^{Agg}$, and two weight lists $w^{Stale}$ and $w^{Volume}$ are updated using the information recorded in the state log $\psi_m$ (lines 2-6). Subsequently, the SoftMax function is applied to transform the values in $w^{Stale}$ and $w^{Volume}$ into the 0-1 range (lines 7). Following this, the weights from $w^{Stale}$ and $w^{Volume}$ are utilized to derive the actual weights $w$ for model aggregation. Similarly, the weights $w$ are mapped to the 0-1 range using the SoftMax function (lines 8-11). Finally, model aggregation is performed on the models within $\Theta^{Agg}$ using the weights from $w$ to formulate the model $\theta$ for the subsequent local training (line 12).

Table 5: Analysis on model aggregation.

| Method | CIFAR10(%) | CIFAR100(%) |
|---|---|---|
| With Average Weights | 90.64 | 73.79 |
| With Only Data Volume Weights $w^{Volume}$ | 91.08 | 73.72 |
| Staleness-aware Aggregation | **91.46** | **74.49** |

Furthermore, we compared our algorithm with the frequently used aggregation methods in FSSL. As shown in Table 5, the staleness-aware model aggregation algorithm demonstrates superior results than the baselines, revealing that our algorithm effectively solves the problem of local models being too stale compared to the received models when aggregating models in the DSSL scenario.

## A.5 System Settings of Experiments

Table 6: Default Decentralized System Settings.

| Decentralized System | Value |
|---|---|
| Number of Clients | 100 |
| Network Connectivity | 0.15 |
| Pre-training Steps | 200 |
| Local Training Epochs | 5 |
| Number of Participants per Step (Parallel Training) | 5 |
| Fine-tuning Epochs | 100 |
| Client Selection Upper Limit | 3 |
| Staleness Bound (Parallel Training) | 5 |
| Depth of Downstream Model | 5 |

Table 7: Default Federated System Settings.

| Federated System | Value |
|---|---|
| Number of Clients | 100 |
| Pre-training Rounds | 200 |
| Local Training Epochs | 10 |
| Number of Participants per Round | 5 |
| Fine-tuning Epochs | 100 |

## A.6 Hyper-parameter Impact Analysis

The system setting of a DSSL scenario involves many hyper-parameters. Experiments are conducted to comprehensively explore the impact of some hyper-parameters on the training of DeNAV.

Table 8: Impact of $S$ on DeNAV.

|  | CIFAR10(%) | CIFAR100(%) | Pre-train Time(min) |
|---|---|---|---|
| $S = 200$ | 90.64 | 73.51 | **37.6** |
| $S = 400$ | 90.85 | 74.87 | 73.1 |
| $S = 600$ | 91.83 | 74.57 | 111.9 |
| $S = 800$ | 92.16 | 75.04 | 154.9 |
| $S = 1000$ | **92.47** | **75.54** | 194.1 |

### A.6.1 Impact of Total Number of Pre-training Steps $S$

The performance of DeNAV was evaluated for different numbers of pre-training steps. Table 8 demonstrates that as the number of pre-training steps $S$ increases, DeNAV achieves better training performance. However, the total training time also increases in proportion to the increase in the number of steps. This effect is consistent with the number of training rounds on training performance in federated learning.

Table 9: Impact of $K$ on DeNAV.

|  | CIFAR10(%) | CIFAR100(%) | Pre-train Time(min) |
|---|---|---|---|
| $K = 1$ | 88.31 | 70.16 | **10.2** |
| $K = 5$ | 90.64 | 73.51 | 37.6 |
| $K = 10$ | 91.41 | 74.98 | 75.5 |
| $K = 15$ | **91.74** | **74.88** | 109.3 |

### A.6.2 Impact of Number of Local Epochs $K$

The parameter $K$ is the number of iterations for each selected client to train the received model using local data. Table 9 shows that the training time is proportional to the value of $K$, but increasing $K$ improves the training outcome of DeNAV. Besides, Table 9 also shows that the training performance improvement by increasing $K$ converges to a certain value. For DeNAV training, if extra time cost is affordable, setting the number of local epochs to a value between 10 and 15 will achieve optimal training performance.

Table 10: Impact of $C$ on DeNAV.

|  | CIFAR10(%) | CIFAR100(%) | Avg Pre-train Time(min) |
|---|---|---|---|
| $C = 1$ | 90.90 | 73.81 | **36.8** |
| $C = 5$ | 91.48 | 73.44 | 40.1 |
| $C = 10$ | 91.81 | 74.04 | 43.4 |
| $C = 15$ | **91.88** | **75.25** | 46.6 |

### A.6.3 Impact of Number of Simultaneously Trained Models $C$

Both decentralized learning and federated learning involve the hyper-parameter $C$, which is the number of models that clients simultaneously train. Table 10 shows that increasing $C$ increases the average training time per model while improving the training performance of DeNAV. However, the effect of $C$ on training time is much smaller compared to the effect of $K$ on time. The increase in training time is not proportional to the increase in $C$.

### A.6.4 Impact of Network Connectivity $\omega$

The DSSL scenarios used in experiments were simulated using the Erdős-Rényi model. This model requires two parameters: the number of nodes in the network and the network connectivity $\omega$. A

Table 11: Impact of $\omega$ on DeNAV.

|  | CIFAR10(%) | CIFAR100(%) |
|---|---|---|
| $\omega = 0.03$ | 89.17 | 72.01 |
| $\omega = 0.15$ | 90.64 | 73.51 |
| $\omega = 0.75$ | **91.16** | **73.65** |

lower value of $\omega$ corresponds to a lower probability that each node in the generated network is connected to other nodes. By varying $\omega$, we evaluated the performance of DeNAV across different networks. As shown in Table 11, the connectivity of the client network significantly affects DeNAV's training results. Specifically, the training performance of DeNAV deteriorates if the client network is sparse. Conversely, in dense networks, the training performance of DeNAV improves.

Table 12: Impact of $H$ on DeNAV.

|  | CIFAR10(%) | CIFAR100(%) | Pre-train Time(min) |
|---|---|---|---|
| $Z = 1$ | 89.51 | 71.90 | **36.4** |
| $Z = 3$ | 90.64 | 73.51 | 37.6 |
| $Z = 5$ | 91.05 | 74.31 | 45.7 |
| $Z = 7$ | **92.15** | **74.72** | 54.2 |

### A.6.5 IMPACT OF CLIENT SELECTION UPPER LIMIT $Z$

DeNAV's client evaluation formula involves an important hyper-parameter $Z$, determining the maximum allowed selections per client. As shown in Table 12, increasing $Z$ improves the training performance of DeNAV while increasing the training time. Additionally, it is also observed that the performance gap between $Z = 1$ and $Z = 3$ is more pronounced in comparison to the gaps between $Z = 3$ and $Z = 5$, and $Z = 5$ and $Z = 7$, since the condition $Z \geq \frac{S}{|M|}$ is unmet when $Z = 1$, resulting in the too early stop on exploiting good clients. Thus, the value of $Z$ needs to be taken in compliance with the condition.

Table 13: Impact of $\lambda$ on DeNAV.

|  | CIFAR10(%) | CIFAR100(%) |
|---|---|---|
| $\lambda = 1$ | 91.39 | 74.35 |
| $\lambda = 5$ | **91.46** | **74.49** |
| $\lambda = 20$ | 90.64 | 73.51 |
| $\lambda = 50$ | 90.52 | 73.45 |
| $\lambda = 200$ | 90.43 | 73.64 |

### A.6.6 IMPACT OF STALENESS BOUND $\lambda$

When multiple models are simultaneously pre-trained in DeNAV, the staleness bound $\lambda$ is employed to restrict the aggregation of local models that have become excessively stale compared to the global model. The results in Table 13 illustrate that relaxing $\lambda$ makes the training performance of DeNAV worse, but overly tightening $\lambda$ also results in limited or no aggregation between the received model and the local models on the same client during pre-training, thereby hindering optimal training performance. Our experimental results reveal that setting $\lambda = 5$ can deliver favorable training results for DeNAV, but tuning of $\lambda$ is suggested for the best training results.

