# OpenReview forum: "DeNAV: Decentralized Self-Supervised Learning with a Training Navigator"
_ICLR.cc/2024/Conference — ICLR 2024 Conference Withdrawn Submission_

### Official Review · Reviewer_b5rW · 2023-10-30

**Soundness:** 2 fair
**Presentation:** 3 good
**Contribution:** 2 fair
**Rating:** 5
**Confidence:** 4

**Summary:**

In realistic FL, it is difficult to ensure that large-scale clients efficiently communicate with a central server. This work studies an essential scenario of Decentralized Self-Supervised Learning (DSSL) based on decentralized communications, in which only unlabeled data is used during the pre-training stage, and the communication between clients involves only model parameters. This paper proposes a method, Decentralized Navigator (DeNAV), utilizing a lightweight pre-training model, namely the One-Block Masked Autoencoder, with a training navigator to evaluate selection scores for the connected clients and plan the training route, eliminating the reliance on server aggregation.

**Strengths:**

1. The proposed scenario is important for FL. The massive clients may lead to in-efficient communication with central server.
2. This paper is written clearly.

**Weaknesses:**

1. The motivation is not strong enough, due to the lack of literature review. The Gossip learning does not constrain that every client must communication with all neighbors. Some Gossip federated learning works also propose to let clients only communicate with one or several neighbors [1][2].
2. In section 3.1, it seems that the described scnario looks like the continual learning. Specifically, the trained model is communicated and trained across clients. How to guarantee the convergence of this training scheme?
3. The proposed methods seem to be limited in the context of transformer-based model architectures.
4. The proposed selection score (6) seems to be little heuristic. How is this equation derived? Can such a selection score ensure convergence?
5. Experiment settings are not clear. What is the non-IID degree used, i.e. alpha in dirichlet sampling? For CIFAR-100 and Mini-INAT show that IID accuracy is better than non-IID, which seems to be impossible.

[1] MATCHA: Speeding Up Decentralized SGD via Matching Decomposition Sampling. In ICC 2019.
[2] GossipFL: A Decentralized Federated Learning Framework With Sparsified and Adaptive Communication. In TPDS 2022.

**Questions:**

See weaknesses.

---

> ### Author Response · Authors · 2023-11-18
>
> Thanks for taking the time and effort to carefully review our paper and give us suggestions. We highly value each of your comments, and all your questions are addressed point by point:
> * **Q1**. First of all, we appreciate your referral to the two papers. Both papers are relative to our research and will be added into the references of our paper. Next, we want to clarify the motivation of this paper. We note that there is a large number of real-world scenarios where clients are inter-connected, holding significant unlabeled data. Methods that can effectively utilize this data are currently extremely lacking. We aim to address this realistic need and deploy our method on some day, and our main contribution is not solely on innovative communication strategy as other decentralized methods.
> * **Q2**. Yes, one or several models are transferred and trained between clients after training begins. However, our approach is actually different to continual learning because after pre-training, each client will use locally saved models to fine-tune downstream tasks without the need to remotely download the latest model. We appreciate your feedback on the lack of convergence analysis and will add this to the revised paper.
> * **Q3**. In fact, our proposed training framework and client selection algorithm works for non-transformer models as well. We have tried replacing the transformer with ResNet-18 and trained models on clients using contrastive learning. The experimental results were again good, but due to page constraints, these results were not put in the paper. We will add this to the revised paper.
> * **Q4**. Our heuristic client evaluation formula is based on Corollary 1 in Section 3.2. The step-by-step development of the formula is outlined in Section 5. We appreciate your feedback on the lack of convergence analysis and will add this to the revised paper.
> * **Q5**. Yes, the non-IID degree is the parameter $\alpha$ in Dirichlet sampling. We will later add a relevant supplement in Section 6 stating that $\alpha=1e5$ in IID experiments and $\alpha=1e-1$ in non-IID experiments. The experimental results in IID settings are often better than those in non-IID settings, as in IID settings, each client in the scenario contains data from all categories.

---

> > ### Comment · Reviewer_b5rW · 2023-11-20
> > **Thanks for your response**
> >
> > Thanks for your responses. Based on your clarifications on the new model architectures and new nonIID experiments, I'd like to increase my score as 5.

---

### Official Review · Reviewer_rt1q · 2023-11-07

**Soundness:** 2 fair
**Presentation:** 2 fair
**Contribution:** 2 fair
**Rating:** 3
**Confidence:** 4

**Summary:**

The paper presents a decentralized self-supervised learning approach based on pre-training an auto encoder that can then be extended and fine-tuned on downstream tasks. The pre-training includes a client selection approach with heuristically defined utility functions. Experiment results confirm the advantage of the proposed DeNAV algorithm compared to baselines.

**Strengths:**

Self-supervised learning is an important and practical topic. This paper considers it in the decentralized/federated scenario, which is good.

**Weaknesses:**

- Decentralized learning has been widely studied in the literature. The training of auto encoders as in this paper is simply a specific type of decentralized learning, where common decentralized SGD algorithms can be applied. It is not quite clear what is new.
- In the same way, client selection has been widely studied in the context of federated learning, where different client selection algorithms have been proposed with convergence analysis. This paper presents a heuristic client selection method. Its advantage over other existing methods that have more theoretical rigor is not clear.
- There is no theoretical analysis of the overall DeNAV algorithm proposed in this paper. Proposition 1 is too informal as a mathematical claim, since any model can be approximated as a linear model if one allows an arbitrarily high approximation error. It is the bound of the approximation error that is more interesting, but such a bound has not been derived. Theorem 1 seems to be simply a least-squares regression result, which is straightforward. In general, it is not quite clear what is the usefulness of the theory presented in Section 3.2, since it is based on possibly inaccurate linear approximation and the main result is straightforward. It does not show the convergence of the overall algorithm, particularly with the client selection mechanism in Section 5.

**Questions:**

Please clarify the concerns mentioned under weaknesses.

---

> ### Author Response · Authors · 2023-11-18
>
> Thanks for taking the time and effort to carefully review our paper and give us suggestions. We highly value each of your comments, and all your questions are addressed point by point:
> * **Q1**. Our research has three main contributions. Firstly, we propose decentralized self-supervised learning based on real-world needs, and our method is the first work in this direction. Secondly, we are also the first to train transformers with limited resources in the decentralized scenario. Our method thus takes advantage over other methods on accuracy. Thirdly, we propose a new training framework that emphasizes training performance, distinguished from other decentralized methods in local training and client selection.
> * **Q2**. We did not compare ours with client selection in federated learning as they are different not only in who initiates the selection but also in the range of selection. Specifically, in federated learning, the server selects the clients that will participate in training from all clients. Whereas in decentralized scenario, each participating client selects a neighbor from the connected clients at the end of training.
> * **Q3**. Corollary 1 in section 3.2 establishes that the pre-training performance is primarily influenced by the data volume on selected clients at each training step, so we prioritize those clients with more data in our client selection. We would like to clarify that not all models can be linear approximated. Our model satisfies linear approximation because of the fixed point, i.e. encoder and decoder will output full-black image after inputting full-black image (0 vectors). Furthermore, we can allow the higher-order approximation error because our focus is on understanding the parameters affecting the optimal training solution rather than giving a rigorous proof of the linear mapping. We appreciate your feedback on the lack of convergence analysis and will add this to the revised paper.